# Zero-Shot Relation Extraction via Reading Comprehension

## Abstract

We show that relation extraction can be reduced to answering simple reading comprehension questions, by associating one or more natural-language questions with each relation slot. This reduction has several advantages: we can (1) learn high-quality relation-extraction models by extending recent neural reading-comprehension techniques, (2) build very large training sets for those models by combining relation-specific crowd-sourced questions with distant supervision, and even (3) do zero-shot learning by extracting new relations that are only specified at test-time, for which we have no labeled training examples. Experiments on a Wikipedia slot-filling task demonstrate that the approach can generalize to new questions for known relations with high accuracy, and that zero-shot generalization to unseen relations is possible, at lower accuracy levels, setting the bar for future work on this task.

## 1 Introduction

Relation extraction systems populate knowledge bases with facts (relations) from an unstructured text corpus. When the type of facts (schema) are predefined, one can use crowdsourcing (Liu et al., 2016) or distant supervision (Hoffmann et al., 2011) to collect examples and train an extraction model for each relation. However, these approaches are incapable of extracting relation types that were *not* specified in advance and observed during training. In this paper, we propose an alternative approach for relation extraction, which can potentially extract relations of new types that were neither specified nor observed a priori.

| Relation | Question Template |
|---|---|
| $educated\_at(x,y)$ | Where did $x$ graduate from? <br> In which university did $x$ study? <br> What is $x$'s alma mater? |
| $occupation(x,y)$ | What did $x$ do for a living? <br> What is $x$'s job? <br> What is the profession of $x$? |
| $spouse(x,y)$ | Who is $x$'s spouse? <br> Who did $x$ marry? <br> Who is $x$ married to? |

Figure 1: Common knowledge-base relations defined by natural-language question templates.

We show that it is possible to reduce relation extraction to the problem of answering simple reading comprehension questions. We map each relation $R(x,y)$ to at least one parametrized natural-language question $q_x$ whose answer is $y$. For example, the relation $educated\_at(x,y)$ can be mapped to "Where did $x$ study?" and "Which university did $x$ graduate from?". Given a particular entity $x$ ("Turing") and a text that mentions $x$ ("Turing obtained his PhD from Princeton"), a non-null answer to any of these questions ("Princeton") asserts the relation and also fills the slot $y$. Figure 1 illustrates a few more examples.

This reduction enables new ways of framing the learning problem. In particular, it allows us to perform *zero-shot learning*: define new relations "on the fly", *after* the model has already been trained. More specifically, the zero-shot scenario assumes access to labeled data for $N$ relations. This data is used to train a reading comprehension model through our reduction. However, at test time, we are asked about a previously unseen relation $R_{N+1}$. Rather than providing labeled data for the new relation, we simply list questions that define the relation's slot values. Assuming a good reading comprehension model has been learned, the correct values should be extracted.

Our zero-shot setup includes innovations both

in data and models. We use distant supervision for a relatively large number of relations (120) from Wikidata (Vrandečić, 2012), which are easily gathered in practice via the WikiReading dataset (Hewlett et al., 2016). We introduce a crowd-sourcing approach for gathering and verifying the questions for each relation. This process produced about 10 questions per relation on average, yielding a dataset of over 30,000,000 question-sentence-answer examples in total. Because questions are paired with relations, not specific examples, this overall procedure has very modest costs.

The key modeling challenge is that most existing reading comprehension problem formulations assume the answer to the question is always present in the given text. However, for relation extraction, this premise does not hold, and the model needs to reliably determine when a question is not answerable. We show that a recent state-of-the-art neural approach for reading comprehension (Seo et al., 2016) can be directly extended with variables to model answerability, and trained directly on our new dataset. This modeling approach is another key advantage of our reduction: as machine reading models improve with time, so should our ability to extract relations.

Experiments demonstrate that our approach generalizes to new paraphrases of questions from the training set, while incurring only a minor loss in performance (4% relative F1 reduction). Furthermore, translating relation extraction to the realm of reading comprehension allows us to extract a significant portion of previously unseen relations, from virtually zero to an F1 of 41%. Our analysis suggests that our model is able to generalize to these cases by learning typing information that occurs across many relations (e.g. the answer to "Where" is a location), as well as detecting relation paraphrases to a certain extent. We also find that there are many feasible cases that our model does not quite master, providing an interesting challenge for future work.

## 2 Approach

We consider the slot-filling challenge in relation extraction, in which we are given a knowledge-base relation $R$, an entity $e$, and a sentence $s$. For example, consider the relation $occupation$, the entity "Steve Jobs", and the sentence "Steve Jobs was an American businessman, inventor, and industrial designer". Our goal is to find a set of text spans $A$ in $s$ for which $R(e, a)$ holds for each $a \in A$. In our example, $A = \{businessman, inventor, industrial designer\}$. The empty set is also a valid answer ($A = \emptyset$) when $s$ does not contain any phrase that satisfies $R(e, ?)$. We observe that given a natural-language question $q$ that expresses $R(e, ?)$ (e.g. "What did Steve Jobs do for a living?"), solving the reading comprehension problem of answering $q$ from $s$ is equivalent to solving the slot-filling challenge.

The challenge now becomes one of *querification*: translating $R(e, ?)$ into $q$. Rather than querify $R(e, ?)$ for every entity $e$, we propose a method of querifying the relation $R$. We treat $e$ as a variable $x$, querify the parametrized query $R(x, ?)$ (e.g. $occupation(x, ?)$) as a question template $q_x$ ("What did $x$ do for a living?"), and then instantiate this template with the relevant entities, creating a tailored natural-language question for each entity $e$ ("What did *Steve Jobs* do for a living?"). This process, *schema querification*, is by an order of magnitude more efficient than instance querification because annotating a relation type automatically annotates all of its instances.

Applying schema querification to $N$ relations from a pre-existing relation-extraction dataset converts it into a reading-comprehension dataset. We then use this dataset to train a reading-comprehension model, which given a sentence $s$ and a question $q$ returns a set of text spans $A$ within $s$ that answer $q$ (to the best of its ability).

In the zero-shot scenario, we are given a new relation $R_{N+1}(x, y)$ at test-time, which was neither specified nor observed beforehand. For example, the $deciphered(x, y)$ relation, as in "Turing and colleagues came up with a method for efficiently deciphering the Enigma", is too domain-specific to exist in common knowledge-bases. We then querify $R_{N+1}(x, y)$ into $q_x$ ("Which code did $x$ break?") or $q_y$ ("Who cracked $y$?"), and run our reading-comprehension model for each sentence in the document(s) of interest, while instantiating the question template with different entities that might participate in this relation. Each time the model returns a non-null answer $a$ for a given question $q_e$, it extracts the relation $R_{N+1}(e, a)$.

Ultimately, all we need to do for a new relation is define our information need in the form of a question. Our approach provides a natural-language API for application developers who are interested in incorporating a relation-extraction

| Relation | Question | Sentence & Answers |
|---|---|---|
| *educated_at* | What is **Albert Einstein**'s alma mater? | **Albert Einstein** was awarded a PhD by the **University of Zürich**, with his dissertation titled... |
| *occupation* | What did **Steve Jobs** do for a living? | **Steve Jobs** was an American **businessman**, **inventor**, and **industrial designer**. |
| *spouse* | Who is **Angela Merkel** married to? | **Angela Merkel**'s second and current husband is quantum chemist and professor **Joachim Sauer**, who has largely... |

Figure 2: Examples from our reading-comprehension dataset. Each instance contains a relation $R$, a question $q$, a sentence $s$, and an answer set $A$. The question explicitly mentions an entity $e$, which also appears in $s$. For brevity, answers are underlined instead of being displayed in a separate column.

component in their programs; no linguistic knowledge or pre-defined schema is needed. To implement our approach, we require two components: training data and a reading-comprehension model. In Section 3, we construct a large relation-extraction dataset and querify it using an efficient crowdsourcing procedure. We then adapt an existing state-of-the-art reading-comprehension model to suit our problem formulation (Section 4).

## 3 Dataset

To collect reading-comprehension examples as in Figure 2, we first gather labeled examples for the task of relation-slot filling. Slot-filling examples are similar to reading-comprehension examples, but contain a knowledge-base query $R(e, ?)$ instead of a natural-language question; e.g. *spouse*(Angela Merkel, ?) instead of "Who is Angela Merkel married to?". We collect many slot-filling examples via distant supervision, and then convert their queries into natural language.

**Slot-Filling Data** We use the WikiReading dataset (Hewlett et al., 2016) to collect labeled slot-filling examples. WikiReading was collected by aligning each Wikidata (Vrandečić, 2012) relation $R(e, a)$ with the corresponding Wikipedia article $D$ for the entity $e$, under the reasonable assumption that the relation can be derived from the article's text. Each instance in this dataset contains a relation $R$, an entity $e$, a document $D$, and an answer $a$. We used distant supervision to select the specific sentences in which each $R(e, a)$ manifests. Specifically, we took the first sentence $s$ in $D$ to contain both $e$ and $a$. We then grouped instances by $R$, $e$, and $s$ to merge all the answers for $R(e, ?)$ given $s$ into one answer set $A$.

**Schema Querification** We then conducted two annotation phases on Mechanical Turk to querify relations: collection and verification.

For each relation $R$, we present the annotator with 4 example sentences, where the entity $e$ in each sentence $s$ is masked by the variable $x$. In addition, we underline the extractable answers $a \in A$ that appear in $s$ (see Figure 3). The annotator must then come up with a question about $x$ whose answer, given each sentence $s$, is the underlined span within that sentence. For example, "In which country is $x$?" captures the exact set of answers for each sentence in Figure 3. Asking a more general question, such as "Where is $x$?" might return false positives ("North America" in sentence 2).

Each worker produced 3 different question templates for each example set. For each relation, we sampled 3 different example sets, and hired 3 different annotators for each set. We ran one instance of this annotation phase where the workers were also given, in addition to the example set, the name of the relation (e.g. *country*), and another instance where it was hidden. Overall, this process collected 40 unique question templates on average, out of a potential 54.

In the verification phase, we measure the question templates' quality by sampling additional sentences and instantiating each question template with the example entity $e$. Annotators are then asked to answer the question from the sentence $s$, or mark it as unanswerable; if the annotators' answers match $A$, the question template is valid. We discarded the templates that were not answered correctly in the majority of the examples (6/10).[1]

Overall, we applied schema querification to 178 relations that had at least 100 examples each (accounting for 99.77% of the data), costing roughly $1,250. After the verification phase, we were left with 1,192 high-quality question templates span-

---

[1]We used this relatively lenient measure because many annotators selected the correct answer, but with a slightly incorrect span; e.g. "American businessman" instead of "businessman". We therefore used token-overlap F1 as a secondary filter, requiring an average score of at least 0.75.

(1) The wine is produced in the **X** region of **France**.
(2) **X**, the capital of **Mexico**, is the most populous city in North America.
(3) **X** is an unincorporated and organized territory of **the United States**.
(4) The **X** mountain range stretches across **the United States** and **Canada**.

Figure 3: An example of the annotator's input when querifying the $country(x, ?)$ relation. The annotator is required to ask a question about $x$ whose answer is, for each sentence, the underlined spans.

ning 120 relations.[2] We then join these templates with our slot-filling dataset along relations, instantiating each template $q_x$ with its matching entities. This process yields a reading-comprehension dataset of over 30,000,000 examples, where each instance contains the original relation $R$ (unobserved by the machine), a question $q$, a sentence $s$, and the set of answers $A$ (see Figure 2).

**Negative Examples** Following the methodology of InfoboxQA (Morales et al., 2016), we generate negative examples by matching (for the same entity $e$) a question $q$ that pertains to one relation with a sentence $s$ that expresses another relation. We also assert that the sentence does not contain the answer to $q$. For instance, we match "Who is Angela Merkel married to?" with a sentence about her occupation: "Angela Merkel is a German politician who is currently the Chancellor of Germany". This process generated 2,249,637 negative examples. While this is a relatively naive method of generating negative examples, our analysis shows that about a third of negative examples contain good distractors (see Section 6).

## 4 Model

Given a sentence $s$ and a question $q$, our algorithm either returns an answer span[3] $a$ within $s$, or indicates that there is no answer.

The task of obtaining answer spans to natural-language questions has been recently studied on the SQuAD dataset (Rajpurkar et al., 2016; Xiong et al., 2016; Lee et al., 2016; Wang et al., 2016). In SQuAD, every question is answerable from the text, which is why these models assume that there exists a correct answer span. Therefore, we modify an existing model in a way that allows it to decide whether an answer exists. We first give a high-level description of the original model, and then describe our modification.

---

[2]58 relations had zero questions after verification due to noisy distant supervision and little annotator quality control.

[3]While our setting allows for multiple answer spans per question, our algorithm assumes a single span; in practice, less than 5% of our data has multiple answers.

We start from the BiDAF model (Seo et al., 2016), whose input is two sequences of words: a sentence $s$ and a question $q$. The model predicts the start and end positions $\mathbf{y}^{start}, \mathbf{y}^{end}$ of the answer span in $s$. BiDAF uses recurrent neural networks to encode contextual information within $s$ and $q$ alongside an attention mechanism to align parts of $q$ with $s$ and vice-versa.

The outputs of the BiDAF model are the confidence scores of $\mathbf{y}^{start}$ and $\mathbf{y}^{end}$, for each potential start and end. We denote these scores as $\mathbf{z}^{start}, \mathbf{z}^{end} \in \mathbb{R}^N$, where $N$ is the number of words in the sentence $s$. In other words, $\mathbf{z}_i^{start}$ indicates how likely the answer is to start at position $i$ of the sentence (the higher the more likely); similarly, $\mathbf{z}_i^{end}$ indicates how likely the answer is to end at that index. Assuming the answer exists, we can transform these confidence scores into pseudo-probability distributions $\mathbf{p}^{start}, \mathbf{p}^{end}$ via softmax. The probability of each $i$-to-$j$-span of the context can therefore be defined by:

$$P(a = s_{i...j}) = \mathbf{p}_i^{start}\mathbf{p}_j^{end} \qquad (1)$$

where $\mathbf{p}_i$ indicates the $i$-th element of the vector $\mathbf{p}_i$, i.e. the probability of the answer starting at $i$. Seo et al. (2016) obtain the span with the highest probability during post-processing.

To allow the model to signal that there is no answer, we concatenate a trainable bias $b$ to the end of both confidences score vectors $\mathbf{z}^{start}, \mathbf{z}^{end}$. The new score vectors $\tilde{\mathbf{z}}^{start}, \tilde{\mathbf{z}}^{end} \in \mathbb{R}^{N+1}$ are defined as $\tilde{\mathbf{z}}^{start} = [\mathbf{z}^{start}; b]$ and similarly for $\tilde{\mathbf{z}}^{end}$, where $[;]$ indicates row-wise concatenation. Hence, the last elements of $\tilde{\mathbf{z}}^{start}$ and $\tilde{\mathbf{z}}^{end}$ indicate the model's confidence that the answer has no start or end, respectively. We apply softmax to these augmented vectors to obtain pseudo-probability distributions, $\tilde{\mathbf{p}}^{start}, \tilde{\mathbf{p}}^{end}$. This means that the probability the model assigns to a null answer is:

$$P(a = \emptyset) = \tilde{\mathbf{p}}_{N+1}^{start}\tilde{\mathbf{p}}_{N+1}^{end}. \qquad (2)$$

If $P(a = \emptyset)$ is higher than the probability of the best span, $\arg\max_{i,j \leq N} P(a = s_{i...j})$, then the model deems that the question cannot be answered

from the sentence. Conceptually, adding the bias enables the model to be sensitive to the absolute values of the raw confidence scores $\mathbf{z}^{start}, \mathbf{z}^{end}$. We are essentially setting and learning a threshold $b$ that decides whether the model is sufficiently confident of the best candidate answer span.

While this threshold provides us with a dynamic per-example decision of whether the instance is answerable, we can also set a global confidence threshold $p_{min}$; if the best answer's confidence is below that threshold, we infer that there is no answer. In Section 5.3 we use this global threshold to get a broader picture of the model's performance.

## 5 Experiments

To understand how well our method can generalize to unseen data, we design experiments for unseen entities (Section 5.1), unseen question templates (Section 5.2), and unseen relations (Section 5.3).

**Evaluation Metrics**   Each instance is evaluated by comparing the tokens in the labeled answer set with those of the predicted span.[4] Precision is the true positive count divided by the number of times the system returned a non-null answer. Recall is the true positive count divided by the number of instances that have an answer.

**Hyperparameters**   In our experiments, we initialized word embeddings with GloVe (Pennington et al., 2014), and did not fine-tune them. The typical training set was an order of 1 million examples, for which 3 epochs were enough for convergence. All training sets had a ratio of 1:1 positive and negative examples, which was chosen to match the test sets' ratio.

### 5.1 Unseen Entities

We show that our reading-comprehension approach works well in a typical relation-extraction setting by testing it on unseen entities and texts.

**Setup**   We partitioned our dataset along entities in the question, and randomly clustered each entity into one of three groups: train, dev, or test. For example, examples about Alan Turing appeared only in the training set, while examples regarding Steve Jobs were exclusive to the test set. We then sampled 1,000,000 examples for train, 1,000 for dev,

---

[4]We ignore word order, case, punctuation, and articles ("a", "an", "the"). We also ignore "and", which often appears when a single span captures multiple correct answers (e.g. "United States and Canada").

and 10,000 for test. This partition also ensures that the sentences at test time are different from those in train, since the sentences are gathered from the Wikipedia article of each entity.

**Results**   Our model correctly predicts the vast majority of test instances, yielding 89.44% F1 (87.66% precision, 91.32% recall). This result suggests that the model does indeed generalize to unseen entities. An analysis of 50 erroneous examples shows that 36% of errors can be attributed to annotation errors (chiefly missing entries in Wikidata), and an additional 42% result from inaccurate span selection (e.g. "8 February 1985" instead of "1985"), for which our model is fully penalized. In total, only 18% of our sample were pure system errors, suggesting that our model is performing even better in practice. This result demonstrates that reducing relation extraction to reading comprehension is indeed a viable approach for our Wikipedia slot-filling task.

### 5.2 Unseen Question Templates

We test our method's ability to generalize to new descriptions of the same relation, by holding out a question template for each relation during training.

**Setup**   We created 10 folds of train/dev/test samples of the data, in which one question template for each relation was held out for the test set, and another for the development set. For instance, "What did $x$ do for a living?" may appear only in the training set, while "What is $x$'s job?" is exclusive to the test set. Each split was stratified by sampling $N$ examples per question template ($N = 1000, 10, 50$ for train, dev, test, respectively). This process created 10 training sets of 966,000 examples with matching development and test sets of 940 and 4,700 examples each.

**Comparison System**   To compare this scenario with one where the question templates were previously observed, we replicated the existing test sets and replaced the unseen question templates with templates from the training set. Revisiting our example, we convert test-set occurrences of "What is $x$'s job?" with "What did $x$ do for a living?".

**Results**   Table 1 shows that our approach is able to generalize to unseen question templates. Our system's performance on unseen questions is nearly as strong as for previously observed templates (losing roughly 3.5 points in F1).

|                   | Precision | Recall  | F1      |
|-------------------|-----------|---------|---------|
| Seen Templates    | 86.73%    | 86.54%  | 86.63%  |
| Unseen Templates  | 84.37%    | 81.88%  | 83.10%  |

Table 1: Performance on previously seen vs. unseen question templates. Both results use the same trained model, but *Seen Templates* tests on examples with templates that appeared in the training set, while *Unseen Templates* tests on examples with unobserved templates.

|                   | Precision | Recall  | F1      |
|-------------------|-----------|---------|---------|
| KB Relation       | 19.32%    | 2.54%   | 4.32%   |
| NL Relation       | 40.50%    | 28.56%  | 33.40%  |
| Single Template   | 37.18%    | 31.24%  | 33.90%  |
| Multiple Templates| 43.61%    | 36.45%  | 39.61%  |
| Question Ensemble | 45.85%    | 37.44%  | 41.11%  |

Table 2: Performance on previously unseen relations. *KB Relation* does not contain any linguistic information about the relation, *NL Relation* uses the relation's name as a natural-language expression, *Single Template* observes one template per relation during training, while *Multiple Templates* observes multiple templates per relation during training. *Question Ensemble* uses the trained *Multiple Templates* model, but is provided with 3 templates per instance at test time.

## 5.3 Unseen Relations

Finally, we test our approach in a true zero-shot setting, where the test-time question templates are not only unobserved during training, they describe new, unseen relations.

**Setup** We created 10 folds of train/dev/test samples, partitioned along relations: 84 relations for training, 12 for development, and 24 for test. For example, when the examples of $educated\_at$ are allocated to the test set, none of them appear in the training set. Each split was stratified by sampling $N$ examples per relation ($N = 10000, 50, 500$ for train, dev, test, respectively). This process created 10 training sets of 840,000 examples each with matching development and test sets of 600 and 12,000 examples per fold.

**Comparison Systems** We compare our querification approach to two alternatives. First, we consider using formal knowledge-base relations instead of questions; these are essentially indicators (e.g. $relation1$) that tie training examples of the same relation together. We expect this alternative to fail, since the new relation's description does not contain any tangible information. The second

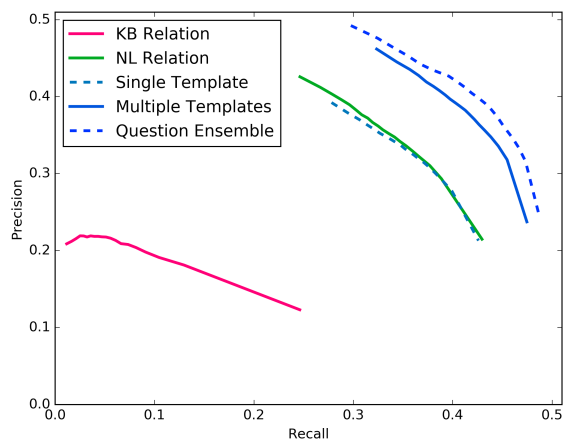

Figure 4: Precision/Recall for unseen relations.

alternative uses the relation's name as a natural-language string instead of a question. For example, $educated\_at$ will be interpreted as "educated at". Unlike the previous baseline, this method allows our machine reading system to interpret its lexical components via word and character embeddings. We also consider a weakened version of our approach where, during training, only one question template per relation is observed. This setting should somewhat hinder the model's ability to learn paraphrases (e.g. "Where" and "Which country", active and passive voice, etc).

**Question Ensemble** We also evaluated how asking (at test time) about the same relation in multiple ways improves performance. This scenario may be more realistic in actual applications, where extractor authors could easily provide a few question paraphrases to define their target relation. We created an ensemble by sampling 3 questions per test example and predicting the answer for each. We then chose the answer with the highest sum of confidence scores.

**Results** Table 2 shows the performance of each setting; Figure 4 extends these results by applying a global threshold on the answers' confidence scores to generate precision/recall curves (see Section 4). As expected, knowledge-base relations are insufficient in a zero-shot setting, and must be interpreted as natural-language to allow for some generalization. The difference between using a single question template and the relation's name appears to be minor. However, training on a variety of question templates rather than on a single expression substantially increases performance. We conjecture that multiple phrasings of the same

| | | |
|---|---|---|
| Verbatim | Relation | András Dombai **plays for** what team? <br> András Dombai... ...currently **plays** as a goalkeeper **for** *FC Tatabánya*. |
| | Type | Which **airport** is most closely associated with Royal Jordanian? <br> Royal Jordanian Airlines... ...from its main base at *Queen Alia International Airport*... |
| Global | Relation | Who was responsible for **directing** Les petites fugues? <br> Les petites fugues is a 1979 Swiss comedy film **directed by** *Yves Yersin*. |
| | Type | **When** was The Snow Hawk released? <br> The Snow Hawk is a *1925* film... |
| Specific | Relation | Who **started** Fürstenberg China? <br> The Fürstenberg China Factory **was founded**... ...**by** *Johann Georg von Langen*... |
| | Type | What **voice type** does Étienne Lainez have? <br> Étienne Lainez... ...was a French operatic *tenor*... |

Figure 5: The different types of discriminating cues we observed among positive examples.

relation allows our model to learn answer-type paraphrases that occur across many relations (see further discussion in Section 6). There is also an advantage to having multiple questions at test time; an ensemble of multiple questions per instance allows for a slight but significant improvement in both precision and recall.

# 6 Analysis

To understand how our method extracts unseen relations, we analyzed 100 random examples, of which 60 had answers in the sentence and 40 did not (negative examples).

For negative examples, we checked whether a distractor – an incorrect answer of the correct answer type – appears in the sentence. For example, the question "Who is John McCain married to?" does not have an answer in "John McCain chose Sarah Palin as his running mate", but "Sarah Palin" is of the correct answer type. We noticed that 14 negative examples (35%) contain distractors. When pairing these examples with the results from the unseen relations experiment in Section 5.3, we found that our method answered 2/14 of the distractor examples incorrectly, compared to only 1/26 of the easier examples. It appears that while most of the negative examples are easy, a significant portion of them are not trivial.

For positive examples, we observed that some instances can be solved by matching the relation in the sentence to that in the question, while others rely more on the answer's type. Moreover, we notice that each cue can be further categorized according to the type of information needed to detect it: (1) when part of the question appears verbatim in the text, (2) when the phrasing in the text deviates from the question in a way that is typical of other relations as well (e.g. syntactic variability), (3) when the phrasing in the text deviates

| | Relation | Type |
|---|---|---|
| Verbatim | 5% | 12% |
| Global | 25% | 8% |
| Specific | 28% | 22% |

Table 3: The distribution of cues by type, based on a sample of 60.

| | Relation | Type |
|---|---|---|
| Verbatim | *33%* | *43%* |
| Global | *73%* | *60%* |
| Specific | 18% | *46%* |

Table 4: Our method's accuracy on subsets of examples pertaining to different cue types. Results in *italics* are based on a sample of less than 10.

from the question in a way that is unique to this relation (e.g. lexical variability). We name these categories *verbatim*, *global*, and *specific*, respectively. Figure 5 illustrates all the different types of cues we discuss in our analysis.

We selected the most important cue for solving each instance. If there were two important cues, each one was counted as half. Table 3 shows their distribution. Type cues appear to be somewhat more dominant than relation cues (58% vs. 42%). Half of the cues are relation-specific, whereas global cues account for one third of the cases and verbatim cues for one sixth. This is an encouraging result, because we can potentially learn to accurately recognize verbatim and global cues from other relations. However, our method was only able to exploit these cues partially.

We paired these examples with the results from the unseen relations experiment in Section 5.3 to see how well our method performs in each category. Table 4 shows the results for the *Multiple Templates* setting. On one hand, the model appears agnostic to whether the relation cue is verbatim, global, or specific, and is able to correctly answer these instances with similar accuracy (there

is no clear trend due to the small sample size). For examples that rely on typing information, the trend is much clearer; our model is much better at detecting global type cues than specific ones.

Based on these observations, we think that the primary sources of our model's ability to generalize to new relations are: *global type detection*, which is acquired from training on many different relations, and *relation paraphrase detection* (of all types), which probably relies on its pre-trained word embeddings.

## 7 Related Work

**Zero-Shot Relation Extraction**  There has been relatively little work on zero-shot learning in information extraction. Rocktäschel et al. (2015) and Demeester et al. (2016) show that injecting precomputed inference rules between natural-language relations and knowledge-base relations in a universal-schema setting (Riedel et al., 2013), allows their system to predict whether a given entity pair takes part in an unseen knowledge-base relation. In their setting, all natural-language relations are observed at train time, while all knowledge-base relations are hidden until test time. This setting is akin to our unseen question templates experiment (Section 5.2) since a natural-language description of each target relation appears in the training data. The "pure" zero-shot scenario – in which no manifestation of the test relation was observed – is substantially more challenging (see Section 5.3).

Bronstein et al. (2015) offered an almost unsupervised approach for event-trigger identification. In their setting, a few seed trigger words are given at test time, and by using a collection of lexical inference and similarity features (tuned by different event types during training), their system is able to detect the desired event-type. We focus instead on the complementary challenge of slot filling, where natural-language questions align perhaps more directly with the values we hope to recover.

Open information extraction (Mausam et al., 2012) can be seen as a type of zero-shot extraction, since there is no need for relation-specific training data. However, such systems often treat every possible string as a new relation while we hope to, through the reduction to reading comprehension, extract a canonical slot value independent of how the original text is phrased.

**Reading Comprehension**  Many reading comprehension formulations have been proposed recently (Richardson et al., 2013; Berant et al., 2014; Hermann et al., 2015; Weston et al., 2015; Hill et al., 2015; Rajpurkar et al., 2016). A typical example involves a text, a question (or cloze), and assumes that the answer is mentioned in the text. In comparison, our questions are relatively direct and easy, but we must also predict when the question is not answerable, to support relation extraction.

**Querification**  Some recent question-answering datasets were collected by expressing knowledge-base assertions in natural language. The Simple QA dataset (Bordes et al., 2015) was created by annotating questions about individual Freebase facts (e.g. $educated\_at(Turing, Princeton)$), collecting roughly 100,000 natural-language questions to support QA against a knowledge graph. Morales et al. (2016) used a similar process to collect questions from Wikipedia infoboxes, yielding the 15,000-example InfoboxQA dataset.

For the task of identifying predicate-argument structures, QA-SRL (He et al., 2015) was proposed as an open schema for semantic roles, in which the relation between an argument and a predicate is expressed as a natural-language question containing the predicate ("Where was someone educated?") whose answer is the argument ("Princeton"). The authors collected about 19,000 question-answer pairs from 3,200 sentences.

In these efforts, the costs scale linearly in the number of instances, requiring significant investments for large datasets. In contrast, schema querification can generate an enormous amount of data for a fraction of the cost by labeling at the relation level; in this work, we generated 30,000,000 examples with a budget of $1,250 (see Section 3).

## 8 Conclusion

We showed that relation extraction can be reduced to a reading comprehension problem, allowing us to generalize to unseen relations that are defined on-the-fly in natural language. However, the problem of zero-shot relation extraction is far from solved, and poses an interesting challenge to both the information extraction and machine reading communities. As research into machine reading progresses, we may find that more tasks can benefit from a similar approach. We hope the contributions and insights presented in this paper will support future work in this avenue.

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
