# Peer review of "Zero-Shot Relation Extraction via Reading Comprehension"

_ACL 2017 — decision unknown_

[Official Review · Reviewer 1 · rating 2 · confidence 4]
soundness 3 · originality 3 · clarity 5 · impact 4 · substance 4 · appropriateness 5 · meaningful comparison 4 · presentation format Poster

- Strengths:
Zero-shot relation extraction is an interesting problem. The authors have
created a large dataset for relation extraction as question answering which
would likely be useful to the community.

- Weaknesses:
Comparison and credit to existing work is severely lacking. Contributions of
the paper don't seen particularly novel.

- General Discussion:

The authors perform relation extraction as reading comprehension. In order to
train reading comprehension models to perform relation extraction, they create
a large dataset of 30m “querified” (converted to natural language)
relations by asking mechanical turk annotators to write natural language
queries for relations from a schema. They use the reading comprehension model
of Seo et al. 2016, adding the ability to return “no relation,” as the
original model must always return an answer. The main motivation/result of the
paper appears to be that the authors can perform zero-shot relation extraction,
extracting relations only seen at test time.

This paper is well-written and the idea is interesting. However, there are
insufficient experiments and comparison to previous work to convince me that
the paper’s contributions are novel and impactful.

First, the authors are missing a great deal of related work: Neelakantan at al.
2015 (https://arxiv.org/abs/1504.06662) perform zero-shot relation extraction
using RNNs over KB paths. Verga et al. 2017 (https://arxiv.org/abs/1606.05804)
perform relation extraction on unseen entities. The authors cite Bordes et al.
(https://arxiv.org/pdf/1506.02075.pdf), who collect a similar dataset and
perform relation extraction using memory networks (which are commonly used for
reading comprehension). However, they merely note that their data was annotated
at the “relation” level rather than at the triple (relation, entity pair)
level… but couldn’t Bordes et al. have done the same in their annotation?
If there is some significant difference here, it is not made clear in the
paper. There is also a NAACL 2016 paper
(https://www.aclweb.org/anthology/N/N16/N16-2016.pdf) which performs relation
extraction using a new model based on memory networks… and I’m sure there
are more. Your work is so similar to much of this work that you should really
cite and establish novelty wrt at least some of them as early as the
introduction -- that's how early I was wondering how your work differed, and it
was not made clear.

Second, the authors neither 1) evaluate their model on another dataset or 2)
evaluate any previously published models on their dataset. This makes their
empirical results extremely weak. Given that there is a wealth of existing work
that performs the same task and the lack of novelty of this work, the authors
need to include experiments that demonstrate that their technique outperforms
others on this task, or otherwise show that their dataset is superior to others
(e.g. since it is much larger than previous, does it allow for better
generalization?)

[Official Review · Reviewer 2 · rating 4 · confidence 4]
soundness 3 · originality 3 · clarity 5 · impact 4 · substance 4 · appropriateness 5 · meaningful comparison 4 · presentation format Oral Presentation

The paper presents a method for relation extraction based on converting the
task into a question answering task. The main hypothesis of the paper is that
questions are a more generic vehicle for carrying content than particular
examples of relations, and are easier to create. The results seem to show good
performance, though a direct comparison on a standard relation extraction task
is not performed.
- Strengths:
The technique seems to be adept at identifying relations (a bit under 90
F-measure). It works well both on unseen questions (for seen relations) and
relatively well on unseen relations. The authors describe a method for
obtaining a large training dataset

- Weaknesses:
I wish performance was also shown on standard relation extraction datasets - it
is impossible to determine what types of biases the data itself has here
(relations are generated from Wikidata via WikiReading - extracted from
Wikipedia, not regular newswire/newsgroups/etc). It seems to me that the NIST
TAC-KBP slot filling dataset is good and appropriate to run a comparison.

One comparison that the authors did not do here (but should) is to train a
relation detection model on the generated data, and see how well it compares
with the QA approach.

- General Discussion:
I found the paper to be well written and argued, and the idea is interesting,
and it seems to work decently. I also found it interesting that the zero-shot
NL method behaved indistinguishably from the single question baseline, and not
very far from the multiple questions system.

[Official Review · Reviewer 3 · rating 2 · confidence 4]
soundness 3 · originality 3 · clarity 4 · impact 4 · substance 3 · appropriateness 5 · meaningful comparison 4 · presentation format Oral Presentation

The paper models the relation extraction problem as reading comprehension and
extends a previously proposed reading comprehension (RC) model to extract
unseen relations. The approach has two main components:

1. Queryfication: Converting a relation into natural question. Authors use
crowdsourcing for this part.

2. Applying RC model on the generated questions and sentences to get the answer
spans. Authors extend a previously proposed approach to accommodate situations
where there is no correct answer in the sentence.

My comments:

1. The paper reads very well and the approach is clearly explained.

2. In my opinion, though the idea of using RC for relation extraction is
interesting and novel, the approach is not novel. A part of the approach is
crowdsourced and the other part is taken directly from a previous work, as I
mention above.

3. Relation extraction is a well studied problem and there are plenty of
recently published works on the problem. However, authors do not compare their
methods against any of the previous works. This raises suspicion on the
effectiveness of the approach. As seen from Table 2, the performance numbers of
the proposed method on the core task are not very convincing. However, this
maybe because of the dataset used in the paper. Hence, a comparison with
previous methods would actually help assess how the current method stands with
the state-of-the-art.

4. Slot-filling data preparation: You say "we took the first sentence s in D to
contain both e and a". How can you get the answer sentence for (all) the
relations of an entity from the first sentence of the entity's Wikipedia
article? Please clarify this. See the following paper. They have a set of rules
to locate (answer) sentences corresponding to an entity property in its
Wikipedia page:

Wu, Fei, and Daniel S. Weld. "Open information extraction using Wikipedia."
Proceedings of the 48th Annual Meeting of the Association for Computational
Linguistics. Association for Computational Linguistics, 2010.

Overall, I think the paper presents an interesting approach. However, unless
the effectiveness of the approach is demonstrated by comparing it against
recent works on relation extraction, the paper is not ready for publication.